# Surgical and Radiological Differences in Intersphenoid Sinus Septation and the Prevalence of Onodi Cells with the Endoscopic Endonasal Transsphenoidal Approach

**DOI:** 10.3390/medicina58101479

**Published:** 2022-10-18

**Authors:** Yun Jin Kang, Il Hwan Lee, Sung Won Kim, Do Hyun Kim

**Affiliations:** 1Department of Otolaryngology-Head and Neck Surgery, Yeouido St. Mary’s Hospital, College of Medicine, The Catholic University of Korea, Seoul 07345, Korea; 2Department of Otolaryngology-Head and Neck Surgery, Chuncheon Sacred Heart Hospital, Hallym University College of Medicine, Chuncheon 24253, Korea; 3Department of Otolaryngology-Head and Neck Surgery, Seoul St. Mary’s Hospital, College of Medicine, The Catholic University of Korea, Seoul 06591, Korea

**Keywords:** skull base, sinus surgery, computed tomography, sphenoid sinus, sphenoid bone

## Abstract

*Background and Objectives*: Understanding the anatomical variation in the sphenoid sinus is important to fully expose the sellar floor and clivus. *Materials and Methods*: The Onodi cell and intersphenoid sinus septation based on preoperative paranasal sinus computed tomography (PNS CT) and the surgical records of 877 patients who underwent the endoscopic endonasal transsphenoidal approach (EETSA) were retrospectively reviewed. *Results*: An intersphenoid sinus septum (ISS) blocking the clivus was defined as a pseudoclivus. Complete and incomplete pseudoclivuses were found in 2.97% and 10.5% of patients, respectively. Intraoperative and PNS CT ISS findings differed in 17.1% of patients. Misconceptions regarding a ridge or vertical ISS and confusion between an incomplete pseudoclivus and a vertical ISS were common. *Conclusions*: Because intraoperative and PNS CT findings may differ, anatomical variation in the paraclival area should be evaluated carefully. A pseudoclivus mimicking the clivus is important to attain a fully exposed EETSA surgical view.

## 1. Introduction

In surgeries that approach the skull base, otolaryngologists and neurosurgeons frequently use the endoscopic endonasal transsphenoidal approach (EETSA) because few complications occur, the tumor-removal results are satisfactory, and the postoperative quality of life of patients is good [1].

The use of the EETSA to remove sellar lesions began in Europe around 1909 [2]. Many studies of the sphenoid sinus have also focused on this surgical approach [3,4]. Subsequently, surgical methods have become more advanced, providing access to not only pituitary lesions but also the cranial fossa, clivus, and planum sphenoidale [5,6].

The varied anatomy of sphenoid sinus structures has been studied and is the most important factor in the EETSA, as it may be associated with postoperative complications or surgical limitations [1,4]. If the internal carotid artery is located very close to the surgical field or if the pneumatization of the sphenoid sinus is insufficient, the EETSA may be difficult [7]. Additionally, complications such as cerebrospinal fluid leaks, endocrinological complications, or carotid artery injury may occur if detailed information, such as on the sphenoid sinus, sellar floor, and paranasal sinus, is not obtained from preoperative radiological images [7].

Variation in the intersphenoid sinus septum (ISS) inside the sphenoid sinus (from the anterior sphenoid wall to the sellar floor and clivus) may also affect the use of an endoscope or the field of vision during surgery [8]. Additionally, the ISS is close to the inner carotid artery and optic nerve, so close attention should be paid in this area [9,10,11]. If the ISS is attached to the internal carotid artery or the optic nerve canal is fractured, it may be an anatomical risk factor for severe bleeding or blindness during EETSA [8]. In addition to ISS, many studies have examined the importance of pneumatization and the sinus dimensions and volume [10,12]. With extensive sphenoid sinus pneumatization, the internal carotid artery or optic nerve canal may dehisce or protrude into the sphenoid sinus. In such cases, iatrogenic damage may result [8]. 

The ISS differs depending on the population [13]. However, this study is important because no study in the English-language literature has analyzed the differences in the ISS between preoperative paranasal sinus computed tomography (PNS CT) and intraoperative findings based on age. An analysis based on age was performed because the variation and ossification of the sphenoid sinus were expected to differ between young and old age groups, as seen in a previous study of sphenoid sinus aeration [14]. We focused on a new concept, a pseudoclivus mimicking the sellar floor.

## 2. Materials and Methods

The surgical records of 877 patients who underwent EETSA-based surgery for a pituitary adenoma, meningioma, Rathke cleft cyst, or other pituitary mass at Seoul St. Mary’s Hospital (the Catholic University of Korea, Seoul, Korea) between February 2009 and July 2020 were reviewed retrospectively. All patients underwent PNS CT. The PNS CT images consisted of 0.6-mm-slice axial projections. First, 0.6-mm-thick coronal and sagittal images were reconstructed under the same non-enhanced bone shadow setting. We focused on Onodi cells and the ISS on PNS CT and on the intraoperative findings. A single surgeon and two assistants evaluated the ISS during EETSA-based surgery, and the location of the Onodi cells was determined in the actual endoscopic view. The Onodi cells are located in the posterior-most ethmoid sinus cells and are superolateral to the sphenoid sinus on PNS CT and in the intraoperative endoscopic view.

All subjects gave informed consent to participate in the study, which was conducted in accordance with the Declaration of Helsinki. The study protocol was approved by the Ethics Committee of the College of Medicine of the Catholic University, Republic of Korea (approval number KC17RESI0354) on 2 November 2017. All data analyses were performed using SPSS 20 (IBM, Armonk, NY, USA). Paired Student’s *t*-tests were used to compare variation in the Onodi cells and ISS between old and young patients. A *p*-value < 0.05 was considered to indicate significance.

### Classification of the Intersphenoid Sinus Septum

In this study, we wanted to confirm whether the classification of the ISS is clinically important for exposing the sellar floor, clivus, and mass. We also wanted to assess the differences between PNS CT findings and actual surgical findings.

The three types of ISS include a vertical ISS, ridge, and pseudoclivus. A vertical ISS was defined as a septum attached to the anterior wall of the sphenoid sinus. A ridge was defined as an ISS that was not attached to the anterior wall of the sphenoid sinus and did not block the sellar floor or clivus. A pseudoclivus was defined as an ISS with vertical and horizontal dimensions in the sphenoid sinus that blocked the clivus, based on axial views on preoperative PNS CT and the actual surgical findings (Figure 1). The pseudoclivus was subclassified as a complete pseudoclivus that was associated with the sphenoid sinus wall or an incomplete pseudoclivus that was not associated with the sphenoid sinus wall. The orientation of all ISSs was defined according to the site of origin as right, left, bilateral, or mid-line [15,16]. Each ISS was classified as having a simple (no ISS, one vertical, or two symmetric vertical ISS) or complex (two or more asymmetrical, three or more of any kind, or the presence of a horizontal ISS) configuration.

## 3. Results

Data from 877 patients (429 males and 448 females) from a retrospective chart review and preoperative PNS CT were analyzed. The mean age of the male patients was 50.7 ± 15.8 years (mean ± standard deviation), that of the female patients was 49.9 ± 15.1 years, and that of all patients was 50.3 ± 15.4 years. Young patients (n = 597) were defined as those < 60 years old and old patients (n = 280) as those > 60 years old. The World Health Organization believes that most developed countries characterize old age from 60 years of age or older.

The various target lesions of EETSA-based surgery are listed in Table 1. Most of the cases underwent EETSA-based surgery for pituitary adenoma (77.7%, particularly pituitary macroadenoma). Meningiomas were detected in 25 (2.9%), craniopharyngiomas in 27 (3.1%), chordomas in 25 (2.9%), and Rathke cysts in 50 (5.7%) patients.

Preoperative CT showed Onodi cells in 449 patients (51.2%). The locations of the Onodi cells are summarized in Table 2. Among the patients with Onodi cells, 301 (34.4%) had bilateral Onodi cells. In total, 371 cases (42.4%) had no Onodi cells. The absence of an Onodi cell was more frequently reported in the young patient group. Onodi cells in the old patient group were located on both the left and right sides or only on the right side. However, no significant differences were observed between young and old patients. The prevalence of an Onodi cell according to the intraoperative findings was the same as that estimated from preoperative PNS CT.

It is important to examine variations in the ISS in advance based on preoperative PNS CT and intraoperative findings. Only 122 cases (13.9%) did not exhibit an ISS. Among the cases with no ISS, more cases with the conchal type were observed in old patients than in young patients (*p* < 0.05; Table 3). Other patients exhibited various types of ISSs (Table 3).

Various types of vertical ISSs were common in most preoperative PNS CT (71.9%) scans. Most of the vertical ISSs were midline vertical ISSs (Figure 2). Bilateral or multiple vertical ISSs with or without a ridge tended to be more common in the older patients, but no significant difference was observed. The prevalence of a complete pseudoclivus with or without a ridge or a vertical ISS on preoperative PNS CT was 2.97% and that of an incomplete pseudoclivus was 10.5%. The presence of a pseudoclivus that was complete on one side and incomplete on the other side was rare (only 0.57%). Significantly more cases of a complete left pseudoclivus with a ridge and only an incomplete right pseudoclivus were observed in old patients. A incomplete right pseudoclivus with a ridge or vertical ISS and an incomplete pseudoclivus with a vertical ISS were found significantly more often in young patients than in old patients. Two pseudoclivuses tended to be more common in young patients on preoperative PNS CT, but no significant differences were detected. A ridge with a pseudoclivus or vertical ISS or a ridge alone was observed in 0.23% of the cases. The ISS was classified as incomplete based on the presence of a ridge. Various types of ISS and pseudoclivus blocking the clivus are shown in Figure 3 and Figure 4.

Most ISSs were midline vertical ISSs based on the intraoperative findings. A left or vertical ISS, a right vertical ISS with a ridge, and a incomplete right pseudoclivus were found significantly more often in young patients than in old patients (Table 3).

Some differences in estimates of the prevalence of an ISS were observed between the actual intraoperative findings and preoperative PNS CT (Figure 5). ISSs were classified differently based on preoperative PNS CT and the intraoperative findings in 17.1% of the cases (150 of 877 cases). Among them, 72% of the cases involved young patients and 28% involved old patients. In most cases, preoperative PNS CT showed a ridge or vertical ISS that was not identified intraoperatively. The next most common cases involved confusion between an incomplete pseudoclivus and a vertical ISS. A complete pseudoclivus was mistaken for an incomplete pseudoclivus and vice versa in a few cases, and the orientation of blockage of the clivus was inaccurately recorded. The complex ISS configuration was slightly more common in young than in old patients.

## 4. Discussion

The EETSA is a very useful surgical method for accessing sellar, parasellar, and suprasellar tumors [5,6]. Understanding the ISS is essential when using this surgical approach. It is critical to determine the type of ISS present on preoperative CT to expose the sellar floor in the surgical field. It is also essential to identify the anatomical relationship between the ISS and the sellar floor during EETSA-based surgery to expose the field of vision such that it includes the planum sphenoidale, medial opticocarotid recess (OCR), and both lateral sides inferior to the clivus.

A fully exposed view of the sellar floor is limited with an Onodi cell during EETSA-based surgery. It becomes difficult to identify structures, such as the OCR, which is located in the anterosuperior portion of the sellar floor. Therefore, the entire lesion can be safely removed with sufficient sellar floor exposure by preoperatively identifying and removing Onodi cells.

The incidence rates of Onodi cells reported in other studies are 7–14% [13]. Asian studies have reported incidences of 32.7–60% [15,16]. The reason for the variation is the different tools used for evaluation, such as cadavers, the provision of an endoscopic view during surgery, or preoperative PNS CT. In this study, the incidence of Onodi cells was 51.26%, which is higher than that reported in other studies that relied on identification from axial, coronal, and sagittal views on preoperative PNS CT [16].

Onodi cells tended to be absent more often in young patients in our study, although there was no statistical significance. This may have been related to the tendency of younger patients to have better sinus aeration than older patients. With age, physical differences, such as height and weight occur, and the development of the sinuses itself may change [17,18]. In addition, Jun et al. reported that the contraction of the skeletal structure due to aging may be related to differences in maxillary sinus aeration [19]. However, Jasmin et al. found no significant difference in the cross-sectional area of the maxillary sinus caused by tooth loss due to aging [20].

The type of ISS varies. A bony ISS extending from the anterior sphenoid sinus wall to the sellar floor must be attached on the left or right of, or along the midline of, the sellar floor, and its placement can be related to the risk of damage to the internal carotid artery or optic nerve [11]. However, an ISS that deviates laterally is close to the clival port of the internal carotid artery, but no bony dehiscence surrounding the internal carotid artery has been found in our surgical experience.

Several studies have analyzed the ISS pattern, which has not been classified by age. In our study, based on actual findings during EETSA-based surgery, the incidences of two, three, and four or five ISSs were 28.96, 10.82, and 0.57%, respectively. The incidence of an incomplete pseudoclivus based on preoperative PNS CT was 10.5%, that of a complete pseudoclivus was 2.97%, and that of an incomplete/complete pseudoclivus was 0.57%, while the respective incidences based on the intraoperative findings were 11.06, 1.82, and 0.34%.

Dziedzic et al. [11] classified the ISS as incomplete or complete and subdivided it into branched and regular types. The types of ISSs reported in other studies have been summarized. In this study, 32% of patients had two ISSs, 3% had three, 21% had a branched ISS, and 79% had a regular ISS.

Zada et al. [21] classified simple and complex ISSs into vertical and horizontal types. The incidence of two ISSs was 32.1%, which was higher than in the present study (29.73%), and the incidence of three or more multiple ISSs was 5.6%, which was lower than in this study (17.21%). No ISS was found in 9% of the cases, which was lower than in this study (13.94%). The incidence of a midline vertical ISS was 18.5%, lower than that in this study (23.6%), and the incidence of an essentially vertical ISS was 34.2%, more than double the value found here (14.61%). The incidence of horizontal and vertical complex ISSs was 9.6%.

Ossification status may vary with age. However, we found a few significant differences between young and old patients in the conchal type, left complete pseudoclivuses with a ridge, right incomplete pseudoclivuses only or those with one ridge or a vertical ISS, and pseudoclivuses that were incomplete on both sides with a vertical ISS as observed on preoperative PNS CT. Significant differences based on intraoperative findings were observed between the young and old patients only for a left or bilaterally vertical ISS, right vertical ISS with a ridge, and right incomplete pseudoclivuses. It has not been confirmed whether the variation of the ISS differs according to age. In older patients, the conchal type was significantly higher, but the ratio was small, and there was no significant difference between the Onodi cell and the ISS except for a few cases. Although maxillary sinus aeration may differ according to age due to skeletal structure contraction due to aging [19], clinically, it can be expected that there is no significant difference between Onodi cells and ISS according to age.

Differences between the preoperative PNS CT and surgical findings were found in about 18% of the cases. A vertical ISS or ridge might not have been visible because it was too small or was removed during EETSA-based surgery. A few cases with a vertical ISS were observed on preoperative PNS CT, but these were cases of incomplete pseudoclivuses. It is important to expose the clivus because an incomplete pseudoclivus may partially block the clivus, even if PNS CT indicates a vertical ISS. A pseudoclivus can be mistaken for the clivus because it usually completely or incompletely blocks the clivus. The complete removal of a pseudoclivus is required to approach the sellar floor. It is possible to approach the exact location of a mass on the clivus during EETSA-based surgery by confirming the orientation of the pseudoclivus.

For successful EETSA, the surgeon must have excellent knowledge of the anatomical relationships in the sphenoid sinus and a detailed examination of the preoperative CT is critical to avoid an increased risk of complications during surgery [8]. However, we identified many cases in which the preoperative CT and intraoperative ISS findings differed. Therefore, the ISS findings should not be unconditionally dependent on CT alone. In our study, a vertical septum on CT was often mistaken for a pseudoclivus. Although a vertical septum was suspected on preoperative CT and removed with a drill or chisel during EETSA, the bony septum in the horizontal dimension was not completely removed, as it might actually be a pseudoclivus mimicking the clivus. Additionally, there may be a ridge that is not identified on CT. A bony ridge attached to the clivus below the sellar floor is useful during EETSA-based surgery to maintain the sphenoid sinus mucosa after elevation. After removing the mass, the elevated sphenoid sinus mucosa can be repositioned without damage.

If the conformation of the ISS is wrong based on checking only the CT images, the sphenoid sinus, sellar floor, and clivus may not be sufficiently removed during EETSA. The protruding internal carotid artery or optic nerve canals around the posterior wall of the sphenoid sinus may not be visible. When removing a pituitary adenoma, surgeons may not be able to obtain a sufficient field of view due to the remaining pseudoclivus. To avoid misunderstandings regarding the ISS, the surrounding bony septum should be removed completely stepwise within the sphenoid sinus in the endoscopic view. Then, a thin, curved probe can be placed behind the septum to determine whether it is a pseudoclivus or a clivus.

This study analyzed the prevalence of clinically important sphenoid sinus structures in a large population. Some limitations of this study should be discussed. First, in a few cases, preoperative PNS CT was performed on patients who underwent revision EETSA-based surgery or had huge masses. Therefore, it was difficult to identify Onodi cells and the presence of an ISS in these cases. Second, further studies are required to identify differences between young and old patients.

## 5. Conclusions

The incidence rates of Onodi cells and more than two ISSs, based on preoperative PNS CT and intraoperative findings, were higher than in previous studies. The prevalence of pseudoclivuses blocking the clivus was 13.22% based on actual intraoperative observations of 877 patients. We found differences related to sphenoid structures between the preoperative PNS CT and intraoperative findings in about 17.1% of the cases. Of these, most of the cases involved young patients. Structures such as a pseudoclivus need to be verified endoscopically rather than only on preoperative PNS CT. It is important to confirm whether a sphenoid structure is blocking the clivus in the endoscopic view. This study confirmed that there are many cases in which there are differences between PNS CT findings and actual surgical findings. Multiple ISSs or a pseudoclivus blocking the clivus should be carefully evaluated operatively, especially in young patients, even if the preoperative CT has been evaluated.

## Figures and Tables

**Figure 1 medicina-58-01479-f001:**
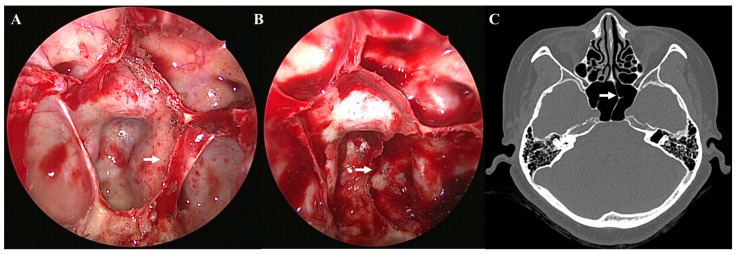
Complete pseudoclivus blocking clivus and left carotid artery. Endoscopic view (**A**) before removal of a left complete pseudoclivus (arrow) blocking clivus and left internal carotid artery (**B**) after the removal of a left complete pseudoclivus with exposed left internal carotid artery. (**C**) Non-enhanced preoperative paranasal sinus computed tomography scan of a patient presenting with a complete left pseudoclivus.

**Figure 2 medicina-58-01479-f002:**
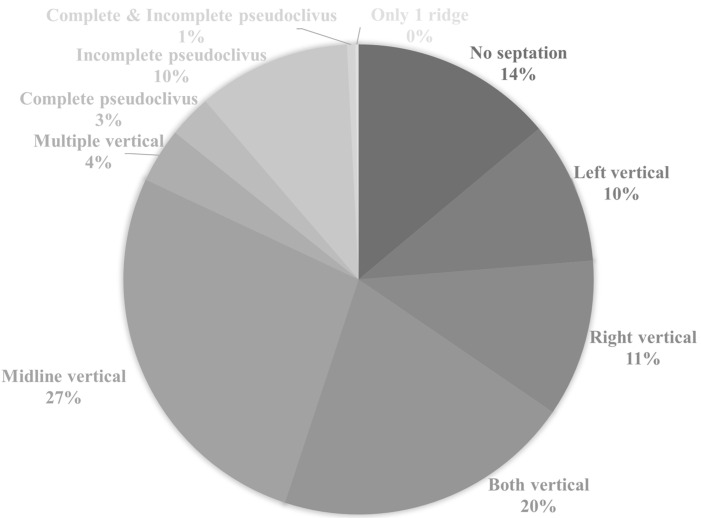
Classification of intersphenoid sinus septation based on preoperative paranasal sinus computed tomography scan. The classification of intersphenoid sinus septation also depends on the presence/absence of another vertical intersphenoid sinus septum or ridge.

**Figure 3 medicina-58-01479-f003:**
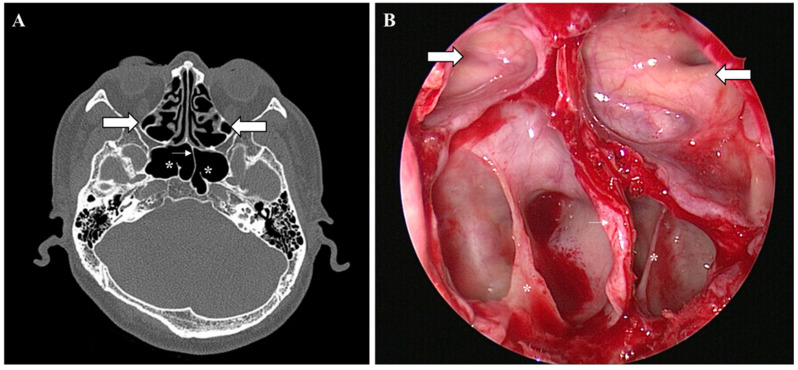
Onodi cell, vertical intersphenoid sinus septum, and ridge. Non-enhanced preoperative paranasal sinus computed tomography scan and endoscopic view of Onodi cell (large arrow), left vertical intersphenoid sinus septation (small arrow), and bilateral ridge (*): (**A**) PNS CT axial view and (**B**) endoscopic view during EETSA-based surgery.

**Figure 4 medicina-58-01479-f004:**
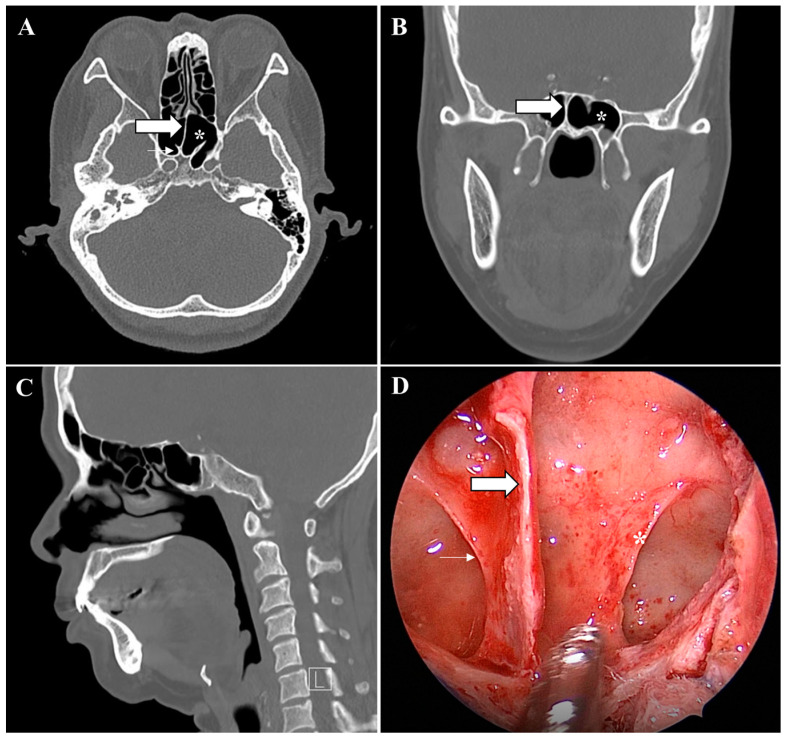
Incomplete pseudoclivus and intersphenoid sinus septum. Non-enhanced preoperative paranasal sinus computed tomography scan and the endoscopic view of the incomplete right pseudoclivus (small arrow), the right vertical intersphenoid sinus septation (large arrow), and the right ridge (*): (**A**) PNS CT axial, (**B**) coronal, (**C**) sagittal, and (**D**) endoscopic view during EETSA-based surgery.

**Figure 5 medicina-58-01479-f005:**
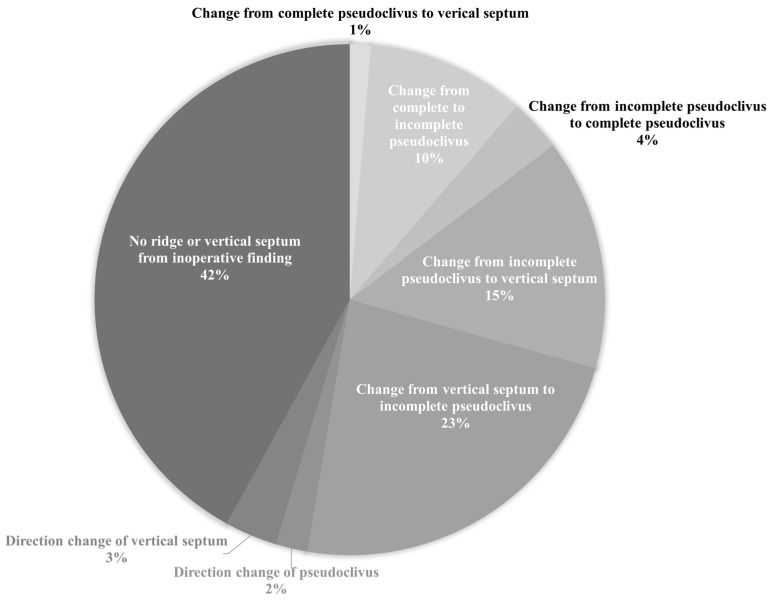
Differences in intersphenoid sinus septation classification based on preoperative paranasal sinus computed tomography scan and intraoperative findings.

**Table 1 medicina-58-01479-t001:** Various lesions were resected using the endoscopic endonasal transsphenoidal approach.

Diagnosis	Number of Patients	Percentage
Pituitary adenoma	681	77.7%
Pituitary apoplexy	9	1.0%
Microprolactinoma	1	0.1%
Meningioma	25	2.9%
Craniopharyngioma	27	3.1%
Chordoma	25	2.9%
Carvenous sinus tumor	2	0.2%
Schwannoma	2	0.2%
Neuroblastoma	3	0.3%
Sarcoma		
- Chondrosarcoma	3	0.3%
- Rhabdomyosarcoma	2	0.2%
- Fibrosarcoma	3	0.3%
Other malignant or metastatic lesions	5	0.6%
Other tumors (myofibroblastic tumor, granular cell tumor, germ cell tumor, lymphoma)	5	0.6%
Cystic lesion		
- Rathke’s cyst	50	5.7%
- Arachenoid cyst	1	0.1%
- Epidermal cyst	2	0.2%
- Pseudocyst	1	0.1%
Hemangioma	5	0.6%
Abscess	2	0.2%
Granuloma	2	0.2%
Fibroma	3	0.3%
Mucocele or pyocele	5	0.6%
Cerebrospinal fluid leakage	2	0.2%
Cushing disease	2	0.2%
Others (mucomycosis, tissue, fibrosis, or fibrous dysplasia)	9	1.0%

**Table 2 medicina-58-01479-t002:** Classification of Onodi cell morphology based on preoperative paranasal sinus computed tomography and intraoperative findings.

Onodi Cell	Total (n = 877)		Young (n = 597)		Old (n = 280)		*p*-Value
	No. of Patients	Percentage	No. of Patients	Percentage	No. of Patients	Percentage	
Right	78	8.90%	50	8.38%	28	10.00%	0.382
Left	70	7.99%	52	8.71%	18	6.43%	0.245
Both	301	34.4%	197	33.00%	104	37.14%	0.228
None	371	42.4%	258	43.22%	113	40.36%	0.485
Not observed due to mass	13	1.48%	8	1.34%	5	1.79%	0.611
Not observed due to previous operation	44	5.02%	33	5.53%	11	3.93%	0.312

**Table 3 medicina-58-01479-t003:** Classification of intersphenoid sinus septation morphology based on the inoperative finding.

ISS	Total (n = 877)		Young (n = 597)		Old (n = 280)		*p*-Value
	No. of Patients	Percentage	No. of Patients	Percentage	No. of Patients	Percentage	
0 septum							
- None	20	2.3%	16	2.68%	4	1.43%	0.247
- Conchal type	2	0.23%	0	0.00%	2	0.71%	0.039 *
- Not observed due to mass	20	2.28%	11	1.84%	9	3.21%	0.205
- Not observed due to previous operation	80	9.13%	58	9.72%	22	7.86%	0.373
Vertical ISS							
- Left	64	7.30%	36	6.03%	28	10.00%	0.035 *
- Left, 1 ridge	17	1.94%	12	2.01%	5	1.79%	0.822
- Left, both ridges	4	0.46%	2	0.34%	2	0.71%	0.437
- Right	78	8.89%	54	9.05%	24	8.57%	0.818
- Right, 1 ridge	14	1.60%	14	2.35%	0	0.00%	0.010 *
- Right, both ridges	3	0.34%	3	0.50%	0	0.00%	0.235
- Both	156	17.79%	95	15.91%	61	21.79%	0.034 *
- Both, 1 ridge	31	3.53%	21	3.52%	10	3.57%	0.968
- Both and midline	26	2.96%	15	2.51%	11	3.93%	0.249
- Both and midline, ridge	5	0.57%	4	0.67%	1	0.36%	0.566
- Midline	207	23.6%	143	23.95%	64	22.86%	0.722
- Midline, 1 ridge	11	1.25%	8	1.34%	3	1.07%	0.739
- Midline, both ridge	18	2.05%	16	2.68%	2	0.71%	0.056
- Left/right, midline	3	0.34%	2	0.34%	1	0.36%	0.958
Complete pseudoclivus							
- Right	6	0.68%	2	0.34%	4	1.43%	0.067
- Right, 1 ridge or vertical	2	0.23%	1	0.17%	1	0.36%	0.583
- Right, 1 vertical & ridge	1	0.11%	0	0.00%	1	0.36%	0.144
- Left	4	0.46%	2	0.34%	2	0.71%	0.437
- Left, 1 ridge or vertical	2	0.23%	1	0.17%	1	0.36%	0.583
- Both	1	0.11%	1	0.17%	0	0.00%	0.493
Incomplete pseudoclivus							
- Right	24	2.74%	21	3.52%	3	1.07%	0.038 *
- Right, 1 ridge/vertical	6	0.68%	4	0.67%	2	0.71%	0.941
- Right, 2 vertical	1	0.11%	0	0.00%	1	0.36%	0.144
- Left	15	1.71%	13	2.18%	2	0.71%	0.119
- Left, 1 ridge/vertical	4	0.46%	2	0.34%	2	0.71%	0.437
- Left, 1 vertical & 1 ridge	9	1.03%	7	1.17%	2	0.71%	0.530
- Both	36	4.10%	28	4.69%	8	2.86%	0.202
- Both, with 1 vertical	2	0.23%	2	0.34%	0	0.00%	0.332
1 Complete, 1 Incomplete pseudoclivus (opposite side)	3	0.34%	2	0.34%	1	0.36%	0.958
Only 1 ridge	2	0.23%	1	0.17%	1	0.36%	0.583

PNS CT: paranasal sinus computed tomography, ISS: intersphenoid sinus septation; * *p* < 0.05 for the test.

## Data Availability

All data generated or analysed during this study are included in this published article.

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
