# Peer review of "Surgical and Radiological Differences in Intersphenoid Sinus Septation and the Prevalence of Onodi Cells with the Endoscopic Endonasal Transsphenoidal Approach"

_medicina, 2022, doi:10.3390/medicina58101479_

Round 1

Reviewer 1 Report

Dr. Kang et al have been compared the diagnostic methods in the intersphenoid sinus septation and tried to find the differences between the methods. However, the contents of manuscript was consisted only to list the observations, no meaningful comparative analysis or quantitative evaluation was performed. No novel findings could be found from the manuscript, as there was no discussion on what could be determined from the obtained results and what kind of considerations could be made from it, and no quantitative analysis of what and how different observation methods differed.

Also, the quality of the manuscript was not sufficient, and the reviewer is unable to fully understand the purpose of this study. In the result section, authors was only described on the observation findings along Tables. The lack of manuscript quality is due to a lack of deeper consideration and further analysis in results and discussion sections. It seems that since the conclusions lacked concreteness, no clarity as to what was obtained from this study.

Reviewer 2 Report

Here are my comments to the text:

1) The english grammatical form of your text is not fluent and is full of errors and inaccurancies. It should be corrected by a proofreader.

2) Line 40: what kind of "postoperative complications" and "surgical limitations" do you mean? Please clarify it.

3) Lines 44-46: for more additional informations about sphenoid sinus pneumatization, risky anatomical variations of sphenoid sinus and the type of attachment of intersphenoidal septum onto internal carotid artery and optic nerve canal, please consider the following work: Fadda GL, Petrelli A, Urbanelli A, Castelnuovo P, Bignami M, Crosetti E, Succo G, Cavallo G. Risky anatomical variations of sphenoid sinus and surrounding structures in endoscopic sinus surgery. Head Face Med. 2022 Sep 3;18(1):29. doi: 10.1186/s13005-022-00336-z. PMID: 36057720; PMCID: PMC9440488.

4) All the text is focused on both Onodi cell and ISS: why Onodi cell does not figure in your title?

5) Line 71: the paragraph is titled "Classification of Onodi cell and ISS" but you don't talk about the classification of Onodi cell, but only of ISS. Please integrate the paragraph with both classifications or rename the paragraph.

6) Line 110: did you find a correlation of this statement in literature? How do you explain yourself that in younger patients Onodi cell is less frequent?

7) According to the possible differences between preoperative CT and intraoperative findings, what kind of solutions do you propose in order to avoid risks or complications during surgery? Please specify.

Round 2

Reviewer 1 Report

Thank you for the response to improve the quality of manuscript.

The authors have made appropriate corrections to address the reviewers' concerns, so there are no further comments from the reviewer.

Author Response

We thank again for your comments and appreciate the previous suggestions to improve the quality of our manuscript. We appreciate this opportunity to respond to their concerns and comments.